# Robustness and Findings of a Web-Based System for Depression Assessment in a University Work Context

**DOI:** 10.3390/ijerph16040644

**Published:** 2019-02-21

**Authors:** Sabina Asensio-Cuesta, Adrián Bresó, Carlos Saez, Juan M. García-Gómez

**Affiliations:** Instituto de Tecnologías de la Información y Comunicaciones (ITACA), Universitat Politècnica de València, Camino de Vera s/n, 46022 Valencia, Spain; adrianbreso@gmail.com (A.B.); carsaesi@ibime.upv.es (C.S.); juanmig@ibime.upv.es (J.M.G.-G.)

**Keywords:** depression, web-based, assessment, Beck Depression Inventory (BDI-II), university, workers

## Abstract

Depression is associated with absenteeism and presentism, problems in workplace relationships and loss of productivity and quality. The present work describes the validation of a web-based system for the assessment of depression in the university work context. The basis of the system is the Spanish version of the Beck Depression Inventory (BDI-II). A total of 185 participants completed the BDI-II web-based assessment, including 88 males and 97 females, 70 faculty members and 115 staff members. A high level of internal consistency reliability was confirmed. Based on the results of our web-based BDI-II, no significant differences were found in depression severity between gender, age or workers’ groups. The main depression risk factors reported were: “Changes in sleep”, “Loss of energy”, “Tiredness or fatigue” and “Loss of interest”. However significant differences were found by gender in “Changes in appetite”, “Difficulty of concentration” and “Loss of interest in sex”; males expressed less loss of interest in sex than females with a statistically significant difference. Our results indicate that the data collected is coherent with previous BDI-II studies. We conclude that the web-based system based on the BDI-II is psychometrically robust and can be used to assess depression in the university working community.

## 1. Introduction

Depression affects a large number of people worldwide [1] and is a considerable burden on both individuals and society, but also in terms of work [2]. At a global level, over 300 million people are estimated to suffer from depression, equivalent to 4.4% of the world’s population [3]. A recent study about the prevalence of depression in the community from 30 countries founds that it was significantly higher in women (14.4%), countries with a medium human development index (HDI) (29.2%), in studies published from 2004 to 2014 (15.4%) and when using self-reporting instruments (17.3%) to assess depression [4]. 

The economic burden of depression, including workplace costs, direct costs and suicide- related costs in the U.S. was estimated to be $210.5 billion in 2010 [5]. In Europe the total cost of depression has been estimated to reach €118 billion, most of which (61%) can be attributed to the indirect costs associated with sick leave and productivity losses. In Spain the economic burden of depression could add up to €5,005 million a year [6].

Stress, depression and anxiety have been reported as the most serious work-related health problem among workers (14%) [7]. Work-related disability and productivity loss in depression are critical determinants of patient quality of life and contribute significantly to the human and economic costs of depression [8]. 4.9% of Spanish workers report suffering from depression or sadness, with a higher percentage of women (6.5%) than men (3.5%). They also report other common depression-related factors, such as: tiredness or exhaustion (18.9%); stress, anxiety or nervousness (17.2%) and sleep problems (9.6%). 60.0% of these workers reported that their depression was either caused or aggravated by their work. It has been estimated that 60% of the visits to the doctor are motivated by depression [9].

Recent research has shown that even experiencing minor depressive symptoms without meeting a clinical diagnosis can lead to impaired health, reduced job performance or absence from work [10], while experiencing depressive symptoms increase the likelihood of a major depression, which may later lead to additional costs, sickness-related absence and suffering [11].

On the other hand, depression, the major cause of suicide, is prevalent but underdetected, under-diagnosed and under-treated, particularly in the case of depressed suicide victims. However, several studies have consistently shown that successful treatment of depression not only relieves depressive symptoms but also reduces suicidality [12].

Among the university community there is a high prevalence of anxiety and depression disorders in university undergraduate-graduate students [13,14], PhD students [15], faculty [16,17,18,19,20] and university staff [21]. This high prevalence in the university community has a detrimental effect on students’ academic performance [22] and on the quality of the teaching imparted [23].

The stigma associated with mental illness can adversely affect help-seeking and employment [24]. Employees are reluctant to disclose their condition to colleagues due to the stigma attached to mental illness [25]. Among students and faculty, a relatively strong stigma exists in regard to depression [26]. People with stigmatizing conditions can benefit from web-based computer education [27] and interventions.

Standard questionnaires are a frequent procedure applied to evaluate the depression severity, such as: Beck Depression Inventory (BDI) [28], Montgomery–Åsberg Depression Rating Scale [29]; Brief Patient Health Questionnaire (PHQ-9) [30]; Zung Self-Rating Depression Scale [31]. These questionnaires have been validated for decades [32]. One of the most widely used is the Beck Depression Inventory (BDI) [28] in its different versions BDI-IA [33] and BDI-II [34]. The BDI-II questionnaire has been adapted to different languages and its psychometric properties have been widely validated from a cross-cultural perspective [32,35,36,37,38,39,40,41,42,43]. Thus, a Spanish version of the BDI-II questionnaire has been validated [38,41,43,44,45,46]. 

On the other hand, in the last years the use of the Internet has extended significantly in the psychological evaluations. The advantages of the internet in relation to online questionnaires are numerous, such as the automated calculation of the result, the consequent saving of time, the reduction of the calculation error, and the possibility of reaching a larger public with low cost [46]. In general, Internet-based psychological treatments seem to be effective for the treatment of depression. While the effects seem to be more favorable for guided or assisted interventions, stand-alone Internet-based treatments for depression have also shown to be effective [47].

Although some studies suggest that questionnaires based on Internet are able to generate equivalent tests paper and pen information in terms of features and psychometric properties [48], the equivalence between the online questionnaires and their original version role cannot be assumed in a general way, and therefore, it is advisable that the adaptations to the Internet be evaluated independently [49].

The International Test Commission (ITC) recommends that when any test type is being adapted to an online format, it is necessary to obtain evidence of equivalence between versions [19].

In summary, Internet-based psychological tests can be reliable and valid, but each questionnaire should be validated for online use [50].

At present, only several studies have validated the BDI-II questionnaire when transferred format from its original paper-and-pencil version to computer version [51] or Internet version [46,47,50,52,53]. Schulenberg et al. [51] concluded that the computerized and paper-and-pencil versions of the BDI-II may be considered equivalent in terms of measurement validity. Carlbring et al. [50] found a significant main effect of administration form for BDI-II. There was a significantly higher score in the Internet version than the paper-and-pencil version, but the effect size was small. Holländare et al. [52] support that the psychometric properties of the BDI-II remained unchanged after transformation to online use. Holländare et al. [46] concluded that the full BDI-II also seems to retain its properties when transferred; however, the item measuring suicidality in the Internet version needs further investigation since it was associated with a lower score in this study. 

The objectives of this study are, first, to validate a web-based system based on the BDI-II questionnaire (Spanish version) [41] to be applied in the university context to assist the prevention of depression, and second, to discuss the findings obtained for the evaluated population.

## 2. Materials and Methods 

An online system based on the BDI-II Spanish questionnaire [41] has been developed [54,55] and it has been added to a website dedicated to preventing depression in the university community to perform this research. 

At the beginning the system shows the user a screen with instructions to perform the questionnaire (identical to those indicated in the paper version of the BDI-II). The questionnaire has been implemented so that each item of the questionnaire is asked in a separate window with the objective of focusing the concentration on the response to each item. The user passes through successive windows until completing the whole questionnaire. In the end, a results window with the overall BDI-II score and corresponding depression level is displayed, as well as the score for each of the 21 items analyzed.

### 2.1. Data Collection

The study has been approved by the ethical committee of the involved university. The participants in the study were recruited face-to-face and by email invitation of the Vice-Rector for Social Responsibility. The invitation included a brief description of what was involved and a link to the BDI-II online questionnaire. All participants that completed the questionnaire were included in the study. 

The study counted with the participation of 185 members of the university staff and faculty. They represented 73% of the departmental teaching and research staff, 100% of the faculties and 46% of the administrative staff’s organizational units. 

Participants were asked to fill the BDI-II online questionnaire with total anonymity. Participants received a brief explanation of the process before filling the questionnaire and received a summary of their results at the end of the session. The online evaluation did not include medical support. The medical support was not included because it was desired to test whether the tool, both from the technical perspective and the questions, was easily understood by the user. The objective of this test was to validate if the use of the tool was possible independently by the user. This would allow it to be used for monitoring and follow-up between patients’ consultations or as a complement to the information collected by the doctor in the consultation. It could also help to reduce the time of consultations since the doctor can have a previous self-assessment that allows a first view of the patient’s situation.

### 2.2. Measurements

The online system developed is based on the widely used Spanish version of the BDI-II [41] approved by the clinical community. 

The BDI-II is composed of 21 items to indicate symptoms related to depression. A total BDI-II score associated with depression severity is calculated from the partial item scores. Minimum and maximum total scores in the test are 0 and 63. Cut-off points exist to classify subjects in one of the groups shown in Table 1. The responses to each item are on a four-point scale, from 0 to 3, except for those of item 16 (changes in sleep patterns), which has seven categories. If a subject chooses various response categories in a single item, the highest-scoring category is taken as the answer.

The online BDI-II assessment considers the following information:The score of each question (total of 21 items), total BDI-II score, corresponding depression level and duration of the test.Demographic information: gender, age.Organizational information:○For faculties: department, school, educational qualifications, work category and seniority.○For staff members: university, unit, type of contract, seniority and type of shift.

### 2.3. Characteristics of the Sample

185 people participated in the study, 88 males and 97 females. Online evaluations were collected for a period of one month. The average age of the participants was 47.93 years; the maximum was 65 and the minimum 32. Table 2 gives the characteristics of the sample and population. Based on these values, the sample can be said to be transversal to the university and representative of the Schools, departments and units. 

The faculty group includes positions organized in three groups: Tenured positions (58.55%), Nontenure positions (41.41%), and others (0.04%). These positions perform researching and teaching activities. The teaching and researching activities are quantified annually by the University with repercussion on economics, promotion and teaching dedication. Moreover, the promotion from one contractual figure to another requires a prior accreditation by an independent agency and a selection process.

The university staff group includes civil servant and contract (permanent and temporary) positions and they are assigned to a department, schools or services. Staff positions profiles and jobs are highly diverse, i.e., administration, management, laboratory technician, computer technician, etc.

## 3. Results

### 3.1. Reliability of the Online BDI-II 

To confirm the online BDI-II reliability in the present study, the internal consistency of the adapted online BDI-II was estimated by the Cronbach coefficient, which was 0.9, indicating an excellent degree of reliability similar to Cronbach coefficient values obtained in previous BDI-II validation studies (which are further discussed and presented in Table 7).

### 3.2. Descriptive Analysis of Online BDI-II Scores

The average overall BDI-II score was 12.7 (+/−1.34; 95% CI), which is within the minimal level of depression but with a standard deviation (SD) of 9.27 that range minimal, mild and moderate levels. The distribution of the scores obtained cannot be regarded as normal at a 5.0% significance level (Shapiro–Wilk test), and shows a positive skewness that implies a large accumulation of individuals around the lower depression level (modal value 8), while the numbers drop steadily until reaching the maximum depression value (see Figure 1). 

### 3.3. Analysis of Online BDI-II Scores by Gender

Of the 185 participants, 97 were female (52.4%) and 88 male (47.5%). The mean female score value was 13.5 (+/−1.86, 95% CI); a value quite close to the mild depression level which starts at 14. The average male score was 11.8 (+/−1.97, 95% CI) (minimal depression level) (see Table 3, Figure 2, Figure 3). There was no statistically significant difference between male and female average scores (α = 5%). 

Given the markedly positive skewness of the scores, the median will be considered as the most robust central indicator. For females, the median was 12 and 10 for males. There is no statistically significant difference between these medians (Mann–Whitney, α = 5%, *p*-value = 0.1637). 

### 3.4. Analysis of Online BDI-II Scores by Age

The average age of the 185 participants was 47.9 (+/-1.03, 95% CI) and the age range was between 32 and 65. The participants’ age distribution followed a normal distribution. There was no significant correlation between age and online BDI-II score (α = 5%; *p*-Value = 0.6881; Coefficient of Correlation = 0.029).

### 3.5. Analysis of Online BDI-II Scores by Work Group 

Of the 185 participants, 115 were from staff (62.1%) and 70 from faculty (37.9%). The average staff score was 12.79 (+/−1.72, 95% CI) and faculty was 12.71 (+/−2.21, 95% CI) (see Table 4, Figure 4 and Figure 5). The staff score range was between 0 and 39, while for faculty it was between 0 and 34. The staff median was 11 and for faculty was 10.5. There was no statistically significant difference between the medians (Mann–Whitney, α = 5%, *p*-value = 0.875).

A Kruskal–Wallis test was performed on staff to find any significant differences in the BDI-II score medians for organizational factors: shift, work category, type of contract and unit (see Table 5). The results found no statistically significant difference between these medians and organizational factors at a 95% confidence level.

Kruskal–Wallis was also applied to faculty scores and organizational factors (see Table 6) and no significant difference was found at a 95% confidence level.

### 3.6. Descriptive Analysis of Online BDI-II Responses (21 items)

The online BDI-II items mean score was 0.60 (+/-0.10705, 95% CI). The responses with the highest values were “16. Changed sleep patterns” (1.19), “15. Loss of Energy” (0.95), “20. Fatigue” (0.81),”12. Loss of Interest” (0.78) (Figure 6). Figure 7 illustrates the BDI-II scores (0, 1, 2, 3) distribution within items.

In relation to the “Changes in sleeping habits” item (see Figure 8) which had the highest mean value. 21.62% of the participants “Experienced no changes in sleeping habits” (score 0); 52.43% slept more or less as before (score 1); 11.35% “slept much more or less than usual” and 14.59% “slept most of the day or woke 1 or 2 hours earlier and could not get back to sleep” (score 3).

Wilcoxon signed-rank tests at a 5% significance level were carried out to test the null hypothesis that there were no statistically significant differences for the different questions between the two principal factors: gender and group (faculty & staff).

Significant differences were found for gender in questions: “18. Changes in appetite” (*p*-value 0.0376), “19. Difficulty of concentration” (*p*-value 0.0376) and “21. Loss of interest in sex” (*p*-value 0.0001). The last question was the one with the greatest gender difference; males expressed less loss of interest in sex than females with a statistically significant difference (Figure 9). 

As regards group (Faculty and Staff) (Figure 10) no statistically significant differences were found for any question the nearest to the rejection of the null hypothesis being “14. Devaluation” with a *p*-value of 0.0756. 

## 4. Discussion

The online BDI-II system reliability is excellent as shown by a Cronbach coefficient of 0.9. This value is similar to previous BDI-II validation studies with the Cronbach coefficient ranged from 0.95 to 0.86 (Table 7).

Table 7 shows BDI-II studies by formats (paper/oral/internet) and samples (adult/adult outpatients/workers) to be compared with the obtained online BDI-II results. The BDI-II mean score in adults/paper ranged 7.7 to 14.2; the online BDI-II mean score (12.7) is within this range. Then the mean score in adults/Internet ranged from 7.3 to 17.89 and the online BDI-II mean score is also within this range. Moreover the online BDI-II mean score is lower than the mean score in adult outpatients/paper (22.1) and it is also lower than mean scores in adult outpatients/Internet (27.4; 31.93), as expected. Related to workers samples the mean score ranged from 8.9 to 14.1 and the online BDI-II mean score (12.7) is also within this range. Therefore we conclude that the online BDI-II developed is able to obtain values within the range of adult/workers samples. 

Depression total scores with the online BDI-II system fall within the ranks of other similar studies carried out to validate the BDI-II (Table 7). Also, a significant correlation exists between the mean scores in our study and the study carried out to validate BDI-II Spanish version in the general population [37]. The system online BDI-II items mean score was 0.60 (+/−0.11, 95% CI), in Sanz et al., (2003) [37] was 0.44 (+/−0.08, 95%CI), the confidence interval for the mean difference was [−1.02; 6.99] that indicates that there is no significant difference between the means of these two data samples, with a confidence level of 95.0%. 

The main depression-associated factors found in the study were: “16. Changed sleep habits” (1.19), “15. Loss of Energy” (0.95), “20. Tiredness or Fatigue” (0.81), “12. Loss of Interest” (0.78), which could be used as indicators for future measures designed to prevent depression in the university. In Sanz et al. (2003) [37] the highest value was also “16. Changes in sleep habits” (0.75) followed by “15. Loss of energy” (0.73), “19. Difficulty concentrating” (0.68) and “20. Tiredness or Fatigue” (0.81). Thus, online BDI-II item’s values for university workers were similar to the previous study related to adult (general population) [37] (Table 8 and Figure 11).

There were no significant differences by gender associated with BDI-II total scores. However significant differences were found by gender in the following items: “18. Changes in appetite”, “19. Difficulty of concentration” and “21. Loss of interest in sex”. The last question was the one with the greatest gender difference; males expressed less loss of interest in sex than females with a statistically significant difference. Thus prevention actions should consider the needs by gender in the context under study.

No significant difference was found in depression levels between the work groups (Staff and Faculty) or in depression-associated problems (21 items). From this it can be concluded that it is not necessary to differentiate between preventive actions for both groups. Nor were significant differences found between organizational factors and depression which would justify transversal preventive policies in the involved university. 

Previous studies have shown that the BDI-II questionnaire retains its properties when transferred to online format [46,47,50,52,53]. In general, the internet offers an opportunity to decrease personal stigma in people with depression widely and at a low cost [24]. In the last decade, a large body of research has demonstrated that internet-based interventions can have beneficial effects on depression. Face-to-face psychotherapeutic interventions for depression can be challenging, so there is a need for alternative methods. Internet-based psychological interventions for depression have been proved to be as beneficial as regular face-to-face therapy [56]. These results open the way to deploying web-based interventions for undergraduate-graduate students, faculty and university staff and provide depression control to the full academic population without increasing the stigma of depression. 

Moreover, technology-free monitoring of patients is time-consuming and expensive due to the need for resources and personnel. The alternative method of using the web-based BDI II presented in this paper could help to monitor patients between consultations or as a complement to the information collected by the doctor face-to-face. It could also help to reduce the time of consultations since the clinician could have a previous patient´s self-assessment as a first view of the problem. Moreover, this tool could contribute to collect a large amount of data about depression in the university to a better knowledge of the problem so that more effective actions in depression prevention, i.e., addressing minor symptoms of depression as prevention.

Furthermore, our study context is a university so that people are used to Internet-based/online technology. Therefore, our results are not transferred to other context involving for instance elder population, in that case, further research to assess the web-based BDI-II in every specific context will be required before its application.

Lastly, we are aware that the study is limited as regards the size of the participant’s sample and the recruitment method. Although the results are promising, larger samples will be needed to continue validating the robustness and results of the web-based BDI-II developed tool. It is important to highlight the possibility of sampling bias due to how participants were recruited.

## 5. Conclusions

Observed indicators in both staff and faculty of depression fell within the ranks of the general population. Besides, the study did not find significant differences between them. Moreover, the study did not find significant differences due to organizational factors. The highest risk factor associated with depression in the university context evaluated was “Changed sleep habits”.

No differences were found in online BDI-II scores as regards age and gender or therefore in depression levels. However, the study found a gender difference in “Changes in appetite”, “Difficulty of concentration” and “Loss of interest in sex” with higher levels in females than male.

This study revealed that the web-based BDI-II developed would be a valid and reliable instrument to assess the presence and severity of depressive symptomatology in university working populations, as shown by many other BDI-II validation studies in other formats (face-to-face, paper), population groups (adults/adult outpatients) and cultures. The online system was found to be reliable for monitoring a large number of users and usable with no technical difficulties were experienced during its use.

## Figures and Tables

**Figure 1 ijerph-16-00644-f001:**
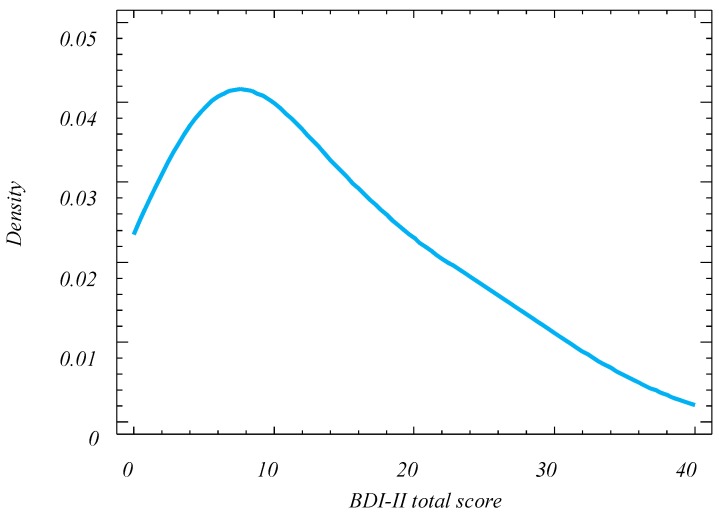
Smoothed density plot of BDI-II scores.

**Figure 2 ijerph-16-00644-f002:**
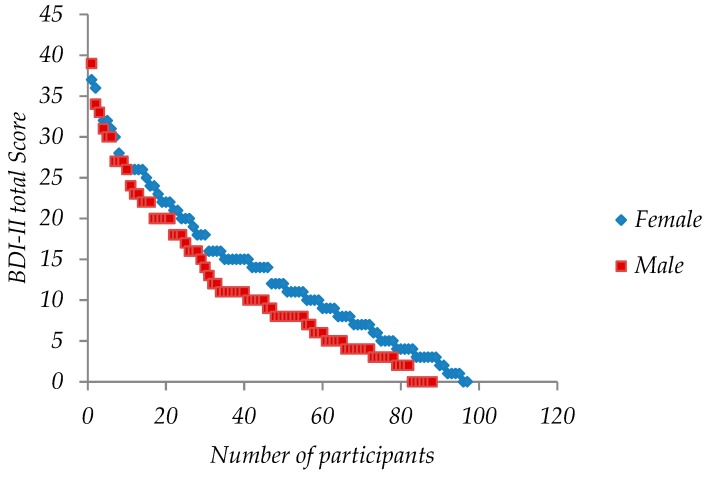
Dispersion plot of BDI-II scores by gender.

**Figure 3 ijerph-16-00644-f003:**
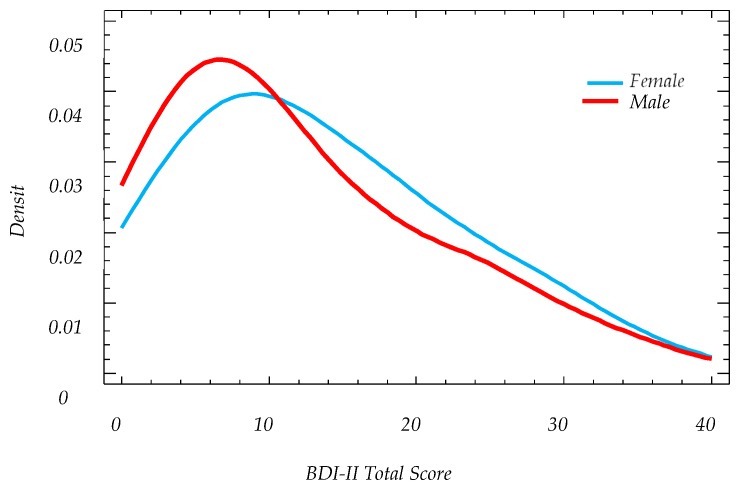
Smoothed density plot of BDI-II scores by gender.

**Figure 4 ijerph-16-00644-f004:**
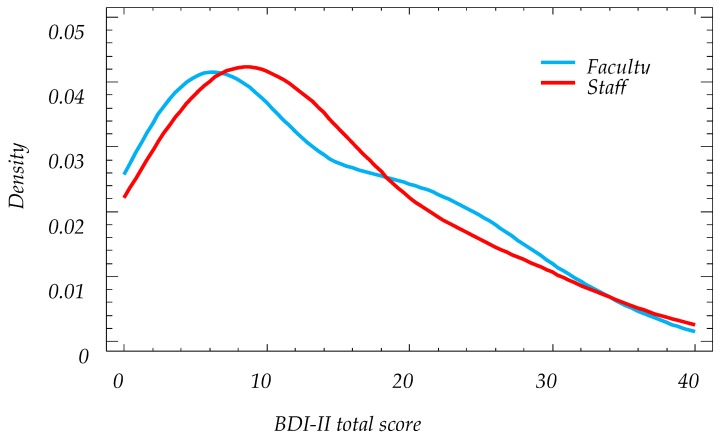
Smoothed density plot of BDI-II scores for staff and faculty.

**Figure 5 ijerph-16-00644-f005:**
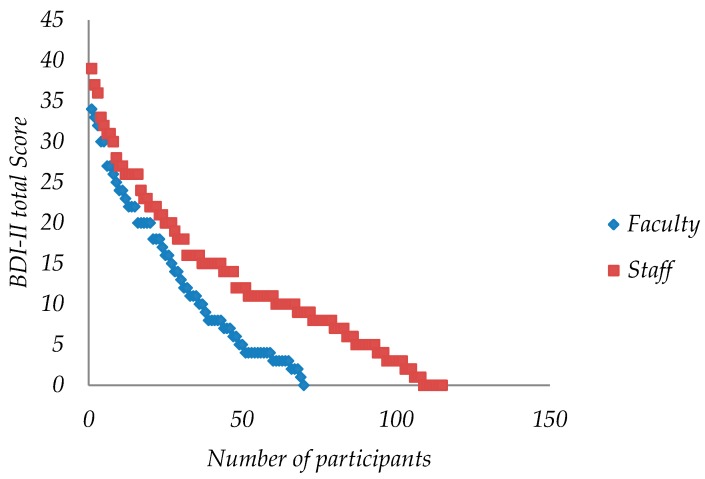
Dispersion plot of BDI-II scores for staff and faculty.

**Figure 6 ijerph-16-00644-f006:**
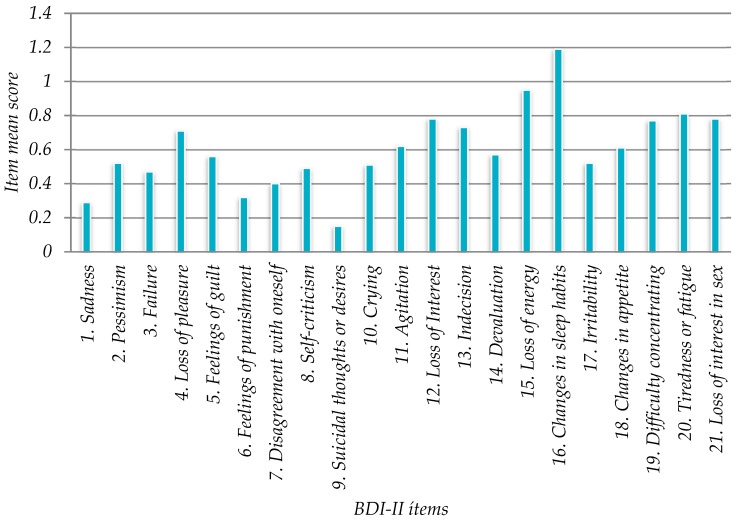
Mean values of 21 BDI-II items.

**Figure 7 ijerph-16-00644-f007:**
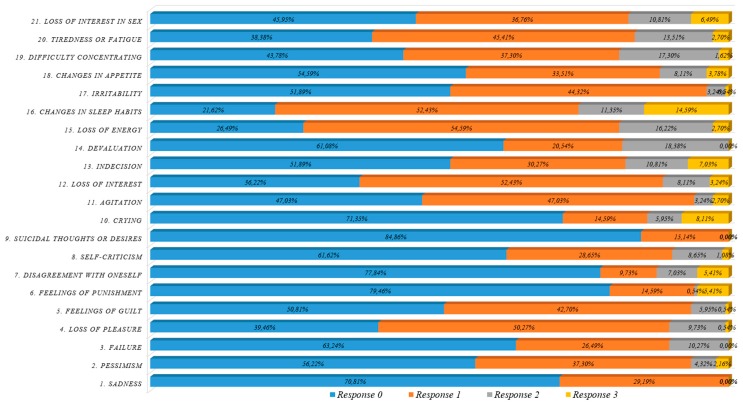
BDI-II questionnaire responses by response level (staff & faculty).

**Figure 8 ijerph-16-00644-f008:**
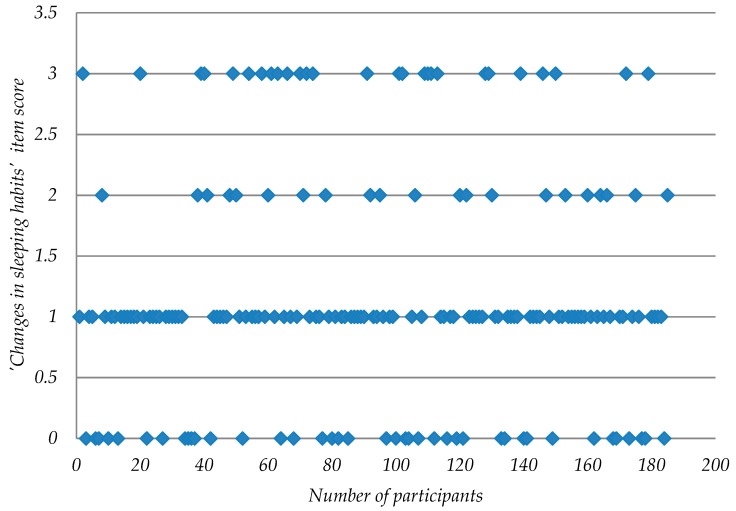
Dispersion plot of BDI-II Item “Changes in sleep habits”.

**Figure 9 ijerph-16-00644-f009:**
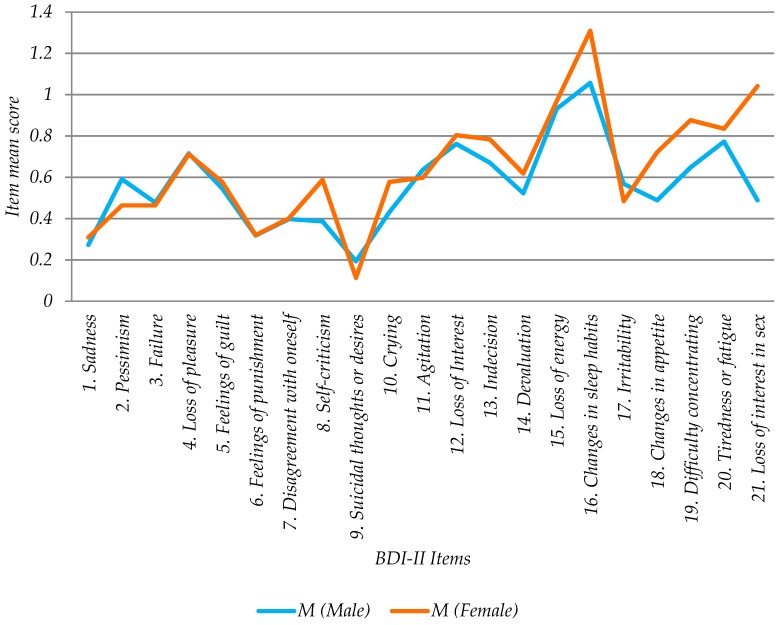
Mean scores (M) of BDI-II responses by gender.

**Figure 10 ijerph-16-00644-f010:**
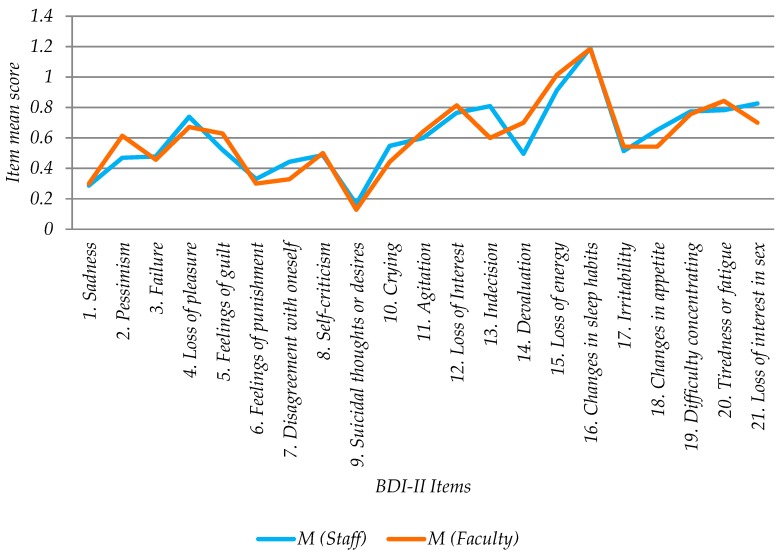
Mean scores (M) of responses by group.

**Figure 11 ijerph-16-00644-f011:**
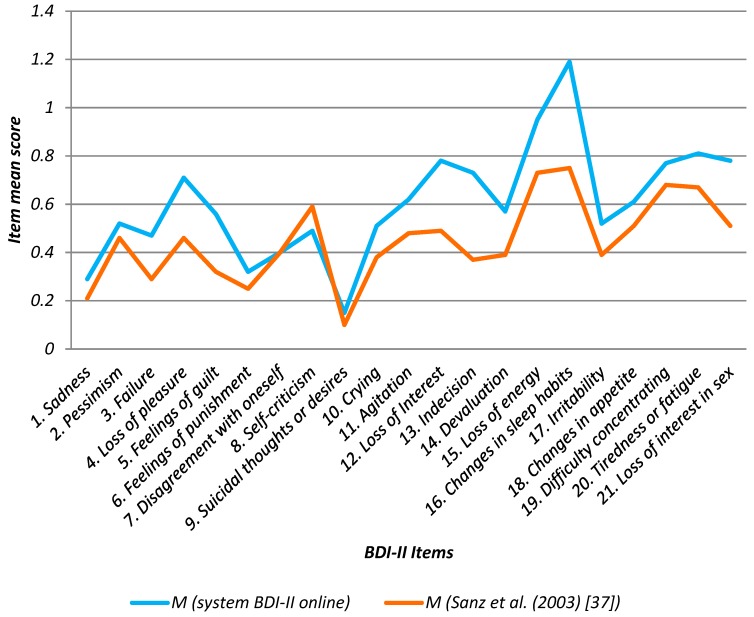
Mean values (M) of 21 BDI-II items in system online BDI-II vs Sanz et al., (2003) [37].

**Table 1 ijerph-16-00644-t001:** Levels of depression defined by the Beck Depression Inventory (BDI) version II (BDI-II). questionnaire.

Level of Depression	Score Range
Minimal	0–13
Mild	14–19
Moderate	20–28
Severe	29–63

**Table 2 ijerph-16-00644-t002:** Population size and sample size and proportion in organizational groups.

Group	Population (#)	Sample (#)	Percentage (%)
Faculty	2641	70	2.5%
Staff	1564	115	7.3%
Departments	42	31	73%
Schools	13	13	100%
Units	121	56	46%

**Table 3 ijerph-16-00644-t003:** Statistical summary of BDI-II scores by gender.

Descriptive Statistics	Females (#)	Males (#)
Count	97	88
Average	13.5	11.8
Standard Deviation	9.2	9.2
Range	37.0	39.0
Standardized Skewness	2.3	3.3

**Table 4 ijerph-16-00644-t004:** Statistical summary of BDI-II scores by group (staff and faculty).

Descriptive Statistics	Staff	Faculty
Count	115	70
Average	12.7913	12.7143
Standard Deviation	9.30996	9.2754
Coefficient of Variation	72.7835%	72.9526%
Minimum	0	0
Maximum	39.0	34.0
Range	39.0	34.0
Standardized Skewness	3.37655	2.10156
Standardized Kurtosis	−0.0490734	−1.27317

**Table 5 ijerph-16-00644-t005:** Kruskal–Wallis analysis of Staff organizational factors.

Factor	Levels	*p* -Value
Shift	8	0.8697
Work Category	6	0.8697
Type of Contract	3	0.1379
Unit	56	0.6083

**Table 6 ijerph-16-00644-t006:** Kruskal-Wallis analysis of faculty organizational factors.

Faculty		
Factor	Levels	*p*-Value
Type of Teaching	7	0.1796
Centre	13	0.8053
Educational Qualifications	8	0.2062
Category	9	0.5139
Department	31	0.6050
Staff		
Factor	Levels	*p*-Value
Shift	8	0.8697
Staff Grade	6	0.8697
Type of Contract	3	0.1379
Unit	56	0.6083

**Table 7 ijerph-16-00644-t007:** BDI-II studies by language version, sample size, target sample, gender distribution, format (paper/Internet), total score mean (standard deviation) and reliability (α) order by format and mean value.

BDI-II study	Language	N	Sample	%Female	Format	Mean	SD	α
(Kühner et al., 2007) [40]	German	89	Adult	51	Paper	7.7	7.5	0.89
(Kojima, 2002) [35]	Japanese	766	Worker	42	Paper	8.9	6.5	0.87
(Sanz et al., 2003) [37]	Spanish	590	Student	78	Paper	9.2	7.5	0.89
(Sanz et al., 2003) [37]	Spanish	470	Adult	53	Paper	9.4	7.7	0.87
(Gomes-Oliveira, 2012) [43]	P. Brasilian	182	Adult	56	Paper	9.9	10.7	0.93
(Aratake, 2007) [30]	Japanese	339	Worker	33	Paper	12.3	8.3	0.90
(Beck et al., 2011) [41]	English	120	Student	56	Paper	12.56	9.93	0.93
(Kapci, 2008) [36]	Turkish	362	Worker	61	Paper	14.1	9.7	0.90
(Ginting et al, 2013) [24]	Indonesia	720	Adult	30	Paper	14.2	9.7	0.86
(Sanz et al., 2005) [38]	Spanish	305	Adult outpatients	75	Paper	22.1	11.5	0.89
(Beck et al., 2011) [41]	English	500	Adult outpatients	63	Paper	22.45	12.75	0.92
(Holländare et al. 2008) [52]	Swedish	71	Student	30	Internet	7.3	7.4	0.94
(Holländare et al. 2008) [52]	Swedish	71	Teacher	30	Internet	9.4	11.1	0.95
Our Online BDI-II	Spanish	185	Worker	52	Internet	12.7	9.2	0.90
(Carlbring et al.2007) [50]	Swedish	350	Adult	49	Internet	17.89	9.6	0.94
(Holländare et al. 2010) [46]	Swedish	43	Adult outpatients	65	Internet	27.4	9.2	0.87
(Holländare et al. 2010) [46]	Swedish	44	Adult outpatients	65	Internet	31.93	10.54	0.89

**Table 8 ijerph-16-00644-t008:** Mean and standard deviation of BDI-II items versus a sample of adult (Sanz et al., 2003) [37].

Item BDI-II	Mean (SD) online BDI-II	Mean (SD) (Sanz et al., (2003) [37]
1. Sadness	0.29 (0.46)	0.21 (0.5)
2. Pessimism	0.52 (0.68)	0.46 (0.7)
3. Failure	0.47 (0.68)	0.29 (0.6)
4. Loss of pleasure	0.71 (0.66)	0.46 (0.7)
5. Feelings of guilt	0.56 (0.63)	0.32 (0.5)
6. Feelings of punishment	0.32 (0.75)	0.25 (0.6)
7. Disagreement with oneself	0.40 (0.84)	0.4 (0.7)
8. Self-criticism	0.49 (0.70)	0.59 (0.7)
9. Suicidal thoughts or desires	0.15 (0.36)	0.1 (0.3)
10. Crying	0.51 (0.93)	0.38 (0.7)
11. Agitation	0.62 (0.68)	0.48 (0.7)
12. Loss of Interest	0.78 (0.73)	0.49 (0.7)
13. Indecision	0.73 (0.92)	0.37 (0.8)
14. Devaluation	0.57 (0.78)	0.39 (0.7)
15. Loss of energy	0.95 (0.73)	0.73 (0.7)
16. Changes in sleep habits	1.19 (0.94)	0.75 (0.8)
17. Irritability	0.52 (0.59)	0.39 (0.6)
18. Changes in appetite	0.61 (0.79)	0.51 (0.7)
19. Difficulty concentrating	0.77 (0.79)	0.68 (0.8)
20. Tiredness or fatigue	0.81 (0.77)	0.67 (0.7)
21. Loss of interest in sex	0.78 (0.88)	0.51 (0.9)

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
