# Peer review of "Robustness and Findings of a Web-Based System for Depression Assessment in a University Work Context"

_ijerph, 2019, doi:10.3390/ijerph16040644_

Round 1

Reviewer 1 Report

Very clear and interesting article. 

Few points:

Can Western Europe be defined more specifically please?

Some further context other than university workplace would be helpful - appreciate that in the data collection section this is mentioned (line 111) but since this article will be of interest to many of those reading it i.e. academics (!) it would be good to have some overall picture of what being an academic is like in Spain.

Why was medical support not included (line 117)?

Line 266 - why is gender capitalised? Also, very interesting findings and some overall gender frameworks or analysis would be great; again to contextualise. i.e. per centre of women/men who are of professorship level?

In terms of conclusions, what relevance of your work (if any) can be used to inform mental health professionals - i.e address minor symptoms of depression as a prevention? Would mental health care professionals be able to access - issues of consent for example; highlighting these aspects rather than in-depth focus would broaden the relevance and application. 

Author Response

Dear Reviewer,
We appreciate your comments to improve our manuscript
Also, English language and style are fine/minor spell has been checked.

Response to Reviewer 1 Comments

Point 1: Can Western Europe be defined more specifically, please?

Response 1: In the introduction, we have included the prevalence of depression across countries instead of merely stating that of Western Europe.

A new paragraph has been added:

“At a global level, over 300 million people are estimated to suffer from depression, equivalent to 4.4% of the world’s population [3]. A recent study about the prevalence of depression in the community from 30 countries founds that it was significantly higher in women (14.4%), countries with a medium human development index (HDI) (29.2%), studies published from 2004 to 2014 (15.4%) and when using self-reporting instruments (17.3%) to assess depression [4].”

Also, the corresponding reference has been included.

“[3] Depression and Other Common Mental Disorders: Global Health Estimates. Geneva: World Health Organization; 2017. Licence: CC BY-NC-SA 3.0 IGO

[4] Lim GY, Tam WW, Lu Y, Ho CS, Zhang MW, Ho, RC. Prevalence of Depression in the Community from 30 Countries between 1994 and 2014. Sci Rep 2018; 8(1): 2861.”

Point 2: Some further context other than university workplace would be helpful - appreciate that in the data collection section this is mentioned (line 111) but since this article will be of interest to many of those reading it i.e. academics (!) it would be good to have some overall picture of what being an academic is like in Spain.

Response 2: Regarding this comment, a new paragraph has been included to clarify the picture of what being an academic (faculty) and staff in a Spanish university.

“The faculty group includes positions organized in three groups: Tenured positions (58, 55%), Non-tenure positions (41, 41%), and others (0, 04%). These positions perform researching and teaching activities. The teaching and researching activities are quantified annually by the University with repercussion on economics, promotion and teaching dedication. Moreover, the promotion from one contractual figure to another requires a prior accreditation by an independent agency and a selection process.

The university staff group includes civil servant and contract (permanent and temporary) positions and they are assigned to a Department, Technical Schools or Services. Staff positions profiles and jobs are highly diverse, i.e. administration, management, laboratory technician, computer technician, etc.”

Point 2: Why was medical support not included (line 117)?

Response 2: the Following paragraph has been included to clarify this point.

“The medical support was not included because it was desired to test whether the tool, both from the technical perspective and the questions, was easily understood by the user. The objective of this test was to validate if the use of the tool was possible independently by the user. This would allow it to be used for monitoring and follow-up between patients' consultations or as a complement to the information collected by the doctor in the consultation. It can also help to reduce the time of consultations since the doctor can have a previous self-assessment that allows a first view of the patient's situation.”

Point 3: Line 266 - why is gender capitalised?

Response 3: We have changed to lowercase.

Point 4: Also, very interesting findings and some overall gender frameworks or analysis would be great; again to contextualise. i.e. per centre of women/men who are of professorship level?

Response 4: We appreciate the reviewer comment. We are planning to analyse depression from a gender perspective in the university context in-depth by applying the web-based BDI-II system presented and validated in this paper. However, we want to highlight that this paper is focused on the assessment of the developed web-based BDI-II more than in prevalence of depression in the university although some general findings are presented.

Point 5: In terms of conclusions, what relevance of your work (if any) can be used to inform mental health professionals - i.e address minor symptoms of depression as prevention? Would mental health care professionals be able to access - issues of consent for example; highlighting these aspects rather than in-depth focus would broaden the relevance and application.

Response 5: To describe the relevance and application of the web-based BDI II presented following paragraph has been included at the end of the discussion section.

“Moreover, technology-free monitoring of patients is time-consuming and expensive due to the need for resources and personnel. The alternative method of using the web-based BDI II presented in this paper could help to monitor patients between consultations or as a complement to the information collected by the doctor face-to-face. It could also help to reduce the time of consultations since the clinician could have a previous patient´s self-assessment as a first view of the problem. Moreover, this tool could contribute to collect a large amount of data about depression in the university to a better knowledge of the problem so that more effective actions in depression prevention, i.e addressing minor symptoms of depression as prevention.”

Reviewer 2 Report

This is a well-written article that is easy and pleasant to read. Some comments are as follows:

1). In the introduction, it is worthwhile to include prevalence of community depression across countries instead of merely stating that of Western Europe. This is especially so when the prevalence of depression may be affected by changes in psychiatric practices and the availability of online mental health information (PMID 29434331). 

2). In the introduction, it is worthwhile to better illustrate how depressive symptoms correlate with disease burden and cost. For instance, in depression, some studies found higher mean Hamilton Rating Scale for Depression score and number of suicide attempts were independent variables associated with increased direct costs while mean Hamilton Rating Scale for Depression scale score was an independent variable for indirect costs. 

3). The authors need to better delineate the recruitment process for online participation. How were participants recruited? Random process?

4). I noted that there was no demographic table for each group of participants. It would be helpful to look at the characteristics of each group and their psychosocial factors. And how many of the participants and proportion in each group were actually depressed? 

5.  There was no mention of strengths and limitations of the study in the discussion. It is important to highlight that the possibility of sampling bias (esp depending on how participants were recruited). How about people/older people who are not comfortable with internet/online technology?

6). The authors did not elucidate convincingly in the introduction/discussion why there is a need for online BDI and not merely giving individuals paper-version of BDI. Also, how would the online and paper version of BDI differ and possibly affect validity? The aim of study is to ascertain online BDI's validity for depression - thus need to explain why there might be a difference between online and paper versions.

7). Some minor grammatical errors, eg line 65 "have" should be "has"; line 102 "is passing successive windows" - sentence structure issue.

Author Response

Dear Reviewer,
We appreciate your comments to improve our manuscript
Also, English language and style are fine/minor spell has been checked.

Response to Reviewer 2 Comments

Point 1: In the introduction, it is worthwhile to include the prevalence of community depression across countries instead of merely stating that of Western Europe. This is especially so when the prevalence of depression may be affected by changes in psychiatric practices and the availability of online mental health information (PMID 29434331). 

Response 1: We agree with this comment and we have changed the Western Europe prevalence sentence to a new paragraph:

At a global level, over 300 million people are estimated to suffer from depression, equivalent to 4.4% of the world’s population [3]. A recent study about the prevalence of depression in the community from 30 countries founds that it was significantly higher in women (14.4%), countries with a medium human development index (HDI) (29.2%), studies published from 2004 to 2014 (15.4%) and when using self-reporting instruments (17.3%) to assess depression [4].

Also, the corresponding reference has been included.

[3] Depression and Other Common Mental Disorders: Global Health Estimates. Geneva: World Health Organization; 2017. Licence: CC BY-NC-SA 3.0 IGO

[4] Lim GY, Tam WW, Lu Y, Ho CS, Zhang MW, Ho, RC. Prevalence of Depression in the Community from 30 Countries between 1994 and 2014. Sci Rep 2018; 8(1): 2861.

Point 2: In the introduction, it is worthwhile to better illustrate how depressive symptoms correlate with disease burden and cost. For instance, in depression, some studies found higher mean Hamilton Rating Scale for Depression score and a number of suicide attempts were independent variables associated with increased direct costs while meaning Hamilton Rating Scale for Depression scale score was an independent variable for indirect costs. 

Response 2: We appreciate the reviewer comment and we will include it in our further investigation regarding online questionaries’ adaptation, such as mentioned Hamilton Rating Scale.

Point 3: The authors need to better delineate the recruitment process for online participation. How were participants recruited? Random process?

Response 3: This paragraph has been included in the Data collection section regarding this comment.

“The participants in the study were recruited face-to-face and by email invitation of the Vice-Rector for Social Responsibility. All participants that completed the questionnaire were included in the study. “

Point 4: I noted that there was no demographic table for each group of participants. It would be helpful to look at the characteristics of each group and their psychosocial factors. And how many of the participants and proportion in each group were actually depressed? 

Response 4: We appreciate the reviewer comment. We are planning to analyse depression prevalence in the university context in-depth by applying the web-based BDI-II system presented and validated in this paper. However, we want to highlight that this paper is focused on the assessment of the developed web-based BDI-II more than in prevalence of depression in the university although some general findings are presented.

Point 5: There was no mention of the strengths and limitations of the study in the discussion. It is important to highlight that the possibility of sampling bias (esp depending on how participants were recruited).

Response 5: This paragraph has been included in the discussion section.

Lastly, we are aware that the study is limited as regards the size of the participant's sample and the participants recruiting process. Although the results are promising, it is important to highlight the possibility of sampling bias due to how participants were recruited.

Point 6: How about people/older people who are not comfortable with internet/online technology?

Response 6: This paragraph has been included in the discussion section

“Furthermore, our study context is a technological university so that people are used to internet/online technology. So, our results are not transferred to other context involving for instance elder population, in that case, further research to assess the web-based BDI-II in every specific context will be required before its application.”

Point 7: The authors did not elucidate convincingly in the introduction/discussion of why there is a need for online BDI and not merely giving individuals paper-version of BDI. Also, how would the online and paper version of BDI differ and possibly affect validity? The aim of the study is to ascertain online BDI's validity for depression - thus need to explain why there might be a difference between online and paper versions.

Response 7: To describe the relevance and application of the web-based BDI II presented following paragraph has been included in the conclusions section.

“Moreover, technology-free monitoring of patients is time-consuming and expensive due to the need for resources and personnel. The alternative method of using the web-based BDI II presented in this paper could help to monitor patients between consultations or as a complement to the information collected by the doctor face-to-face. It could also help to reduce the time of consultations since the clinician could have a previous patient´s self-assessment as a first view of the problem. Moreover, this tool could contribute to collect a large amount of data about depression in the university to a better knowledge of the problem so that more effective actions in depression prevention, i.e addressing minor symptoms of depression as prevention.”

Point 8: Some minor grammatical errors, eg line 65 "have" should be "has"; line 102 "is passing successive windows" - sentence structure issue.

Response 8: We appreciate reviewer corrections; errors have been corrected.

Line 65: Thus, a Spanish version of the BDI-II questionnaire has been validated…

Line 102 change to  “The user passes through successive windows until completing the whole questionnaire.”